# SIMPLE CNN FOR VISION

## ABSTRACT

Traditional Convolutional Neural Networks (CNNs) tend to use 3×3 small kernels, but can only capture neighboring spatial information in one block. Inspired by the success of Vision Transformers (ViTs) in capturing long-range visual dependencies, recent CNNs have reached a consensus on utilizing large kernel convolutions (e.g., 31×31 and, astonishingly, 51×51 kernels). Nevertheless, these approaches necessitate adopting specialized techniques such as re-parameterization or sparsity, which require extra post-processing. And too large kernels are unfriendly to hardware. This paper introduces a Simple Convolutional Neural Network (SCNN) that employs a sequence of stacked 3×3 convolutions but surpasses state-of-the-art CNNs utilizing larger kernels. Notably, we propose simple yet highly effective designs that enable 3×3 convolutions to progressively capture visual cues of various sizes, thereby overcoming the limitations of smaller kernels. First, we build a thin and deep model, which encourages more convolutions to capture more spatial information under the same computing complexity instead of opting for a heavier, shallower architecture. Furthermore, we introduce an innovative block comprising two 3×3 depthwise convolutions to enlarge the receptive field. Finally, we replace the input of the popular Sigmoid Linear Unit (SiLU) activation function with global average pooled features to capture all spatial information. Our SCNN performs superior to state-of-the-art CNNs and ViTs across various tasks, including ImageNet-1K image classification, COCO instance segmentation, and ADE20K semantic segmentation. Remarkably, SCNN outperforms the small version of Swin Transformer, a well-known ViT, while requiring only 50% computation, which further proves that large kernel convolution is not the only choice for high-performance CNNs.

## 1 INTRODUCTION

The field of neural network architecture holds paramount significance within machine learning and computer vision research. In recent years, notable Vision Transformer (ViT) architectures (Dosovitskiy et al., 2021; Touvron et al., 2021) with global attention have been sequentially introduced. These advancements have considerably enhanced the performance of various computer vision tasks and surpassed convolutional neural networks (CNNs) by a large margin.

Very recently, Swin Transformer (Liu et al., 2021b) captures the spatial patterns using local shifted window attention and obtains comparable results with the ViTs using the global window. This local attention is viewed as a variant of the large kernel. Thus, some novel CNNs use large convolutional kernels to improve performance to strike back against the ViTs. DWNet (Han et al., 2022) replaced the local attention in Swin (Liu et al., 2021b) with the 7×7 depthwise convolutional layer and surprisedly found it could obtain the same result on image classification with only 84% computation. Almost at the same time, ConvNeXt (Liu et al., 2022) gradually modernized a standard ResNet toward the design of ViTs and scale the kernel size from 3×3 to 7×7. Following this large kernel design, many advanced CNN-based architectures (Ding et al., 2022; Liu et al., 2023; Yu et al., 2023) were proposed, and they achieved more impressive results in more vision tasks. InceptionNeXt (Yu et al., 2023) decomposed large-kernel depthwise convolution into four parallel branches along channel dimension to design a novel Inception-style CNN (Szegedy et al., 2015; Ioffe & Szegedy, 2015; Szegedy et al., 2016; 2017). RepLKNet (Ding et al., 2022) employed re-parameterized $31 \times 31$ convolutions to build up large receptive fields. SLaK (Liu et al., 2023) introduced a sparse factorized $51 \times 51$ convolution to simplify the training difficulty of a large kernel. However, their results are

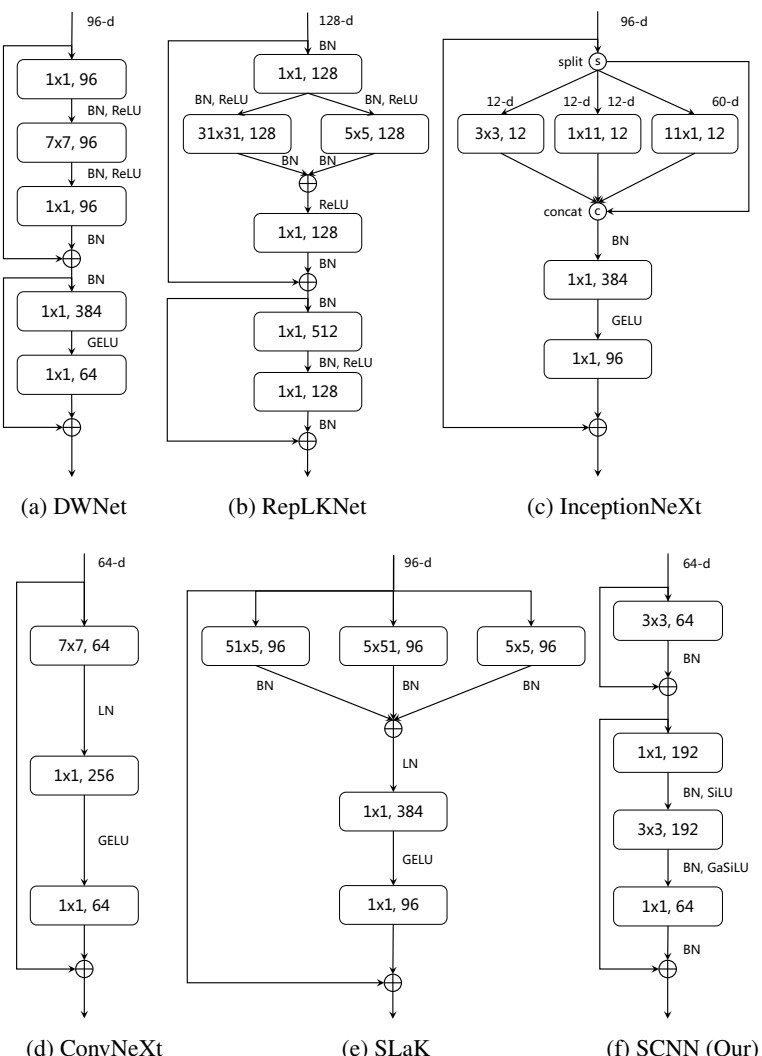

Figure 1: **Block illustration of DWNet, RepLKNet, InceptionNeXt, ConvNext, SLaK and SCNN.** The former five CNNs introduce large kernel, and the smallest kernel size is set to $7 \times 7$. Our SCNN is more efficient because only $3 \times 3$ kernel depthwise convolution is used.

much poorer than recent state-of-the-art ViTs (Dong et al., 2022) and MLPs (Lai et al., 2023). In addition, the large kernel improves training difficulty and is unfriendly to hardware devices. Moreover, some large kernel methods (Ding et al., 2022; Liu et al., 2023) require extra processes and complicated architecture.

Is the large kernel really CNN needed? This paper makes an interesting thing: **stacking depthwise $3 \times 3$ convolutions in a simple CNN architecture** and outperforming (efficiency and effectiveness) state-of-the-art MLPs, CNNs, and ViTs. The overall architecture, Simple CNN (SCNN), is shown in Figure 1f, which consists of pure small kernels. In previous small kernel CNNs (He et al., 2016; Xie et al., 2017; Sandler et al., 2018; Radosavovic et al., 2020; Zhang et al., 2022), researchers focus more on the design of bottleneck block and ignore the importance of the receptive field in the network. In particular, we make some simple but effective designs to let $3 \times 3$ convolutions progressively capture various sizes of visual cues in one block, which breaks through the limitation of small kernels. First, we designed a thin and deep model to capture more spatial information instead of a heavy and shallow one, which could have more $3 \times 3$ convolutions under the same computing complexity. We then introduce a novel block with two $3 \times 3$ depthwise convolutions to

enlarge the receptive field further. Finally, we replace the input of the popular Sigmoid Linear Unit (SiLU) activation function with global average pooled features, which lets SCNN capture global information. Impressively, the overall SCNN architecture is simple but effective and outperforms existing complicated architectures.

Our SCNN achieves the best accuracy in ImageNet-1K image classification compared to state-of-the-art ViTs, MLPs, and CNNs. Moreover, compared with the state-of-the-art CNN, SLaK, our improvements (0.7% accuracy on the tiny scale, same accuracy on the small and base scale with fewer FLOPs) are significant. Also, compared with the well-known Swin Transformer and ConvNeXt, our tiny version could obtain better results with almost 50% of the computation.

SCNN also achieves impressive results and outperforms state-of-the-art CNNs, ViTs, and MLPs on dense prediction tasks, including MS-COCO object detection, MS-COCO instance segmentation, and ADE20K semantic segmentation. In particular, our SCNN outperforms previous state-of-the-art CNNs by a large margin (around 0.9% $Ap^b$ or 0.5% mIoU improvements).

The Above experimental results with our simple architecture demonstrate not only the effectiveness and efficiency of our model but the great potential of CNNs in both image classification and dense prediction. We believe this paper will raise more attention to CNNs for vision. Our contributions can be summarized below:

- We introduce a small kernel CNN architecture named Simple CNN, which employs a thin and deep architecture to capture more spatial information. A novel block with two 3×3 depthwise convolutions is also proposed to enlarge the receptive field of the model further.

- A Global Sigmoid Linear Unit activation function is proposed to capture global visual cues, which leads to richer spatial feature extraction.

- Extensive experiments demonstrate that SCNN outperforms the state-of-the-art CNNs, ViTs, and MLPs in various vision tasks, including image classification, object detection, instance segmentation, and semantic segmentation.

## 2 RELATED WORK

**Convolutional Neural Network Architectures.** The introduction of AlexNet (Krizhevsky et al., 2012) marked a significant milestone in the rapid development of Convolutional Neural Networks (CNNs), with subsequent architectures (Szegedy et al., 2015; He et al., 2016; Szegedy et al., 2017) continually pushing the boundaries of performance. One recent trend in CNNs is the utilization of large convolutional kernels to achieve larger receptive fields and capture more long-range information. ConvNeXt (Liu et al., 2022) has made a noteworthy discovery, revealing that scaling the kernel size from 3×3 to 7×7 significantly contributes to performance. Similarly, DWNet (Han et al., 2022) has reached a similar conclusion by replacing the local attention layer in Swin (Liu et al., 2021b) with a 7×7 depthwise convolutional layer. Additional architectures, such as RepLKNet (Ding et al., 2022) and SLaK (Liu et al., 2023), have also demonstrated impressive outcomes in many vision tasks, employing even larger kernel sizes like 31×31 and 51×51 convolutions, respectively. However, some methods introduce complicated architecture to employ large kernels. In addition, using large kernels in models will improve training difficulty and is hardware-intensive, resulting in longer training and inference times.

**Multi-Layer Perceptron Architectures.** Recently, there has been a surge in the popularity of Multi-Layer Perceptron (MLP)-based architectures. MLP-Mixer (Tolstikhin et al., 2021) applies MLP independently to mix spatial information and per-location information without any spatial convolution and transformer blocks. Building upon this, Res-MLP (Touvron et al., 2022) and gMLP (Liu et al., 2021a) adopts two-layer feed-forward network and Spatial Gating Unit to enhance the performance of pure MLP-based architectures, respectively. Further advancements are made with $S^2$-MLP (Yu et al., 2022), AS-MLP (Lian et al., 2022), and Hire-MLP (Guo et al., 2022). These methods introduced spatial-shift MLP for capturing local and global information. Recently, RaMLP (Lai et al., 2023) presents Region-aware Mixing to capture and capture visual dependence in a coarse-to-fine manner and even outperforms the state-of-the-art CNNs, ViTs. However, MLPs are still difficult to dense prediction tasks, such as object detection and semantic segmentation.

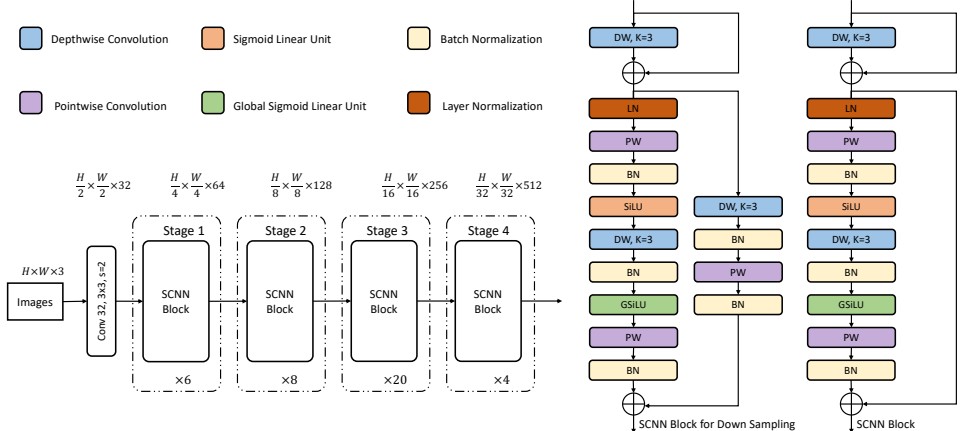

Figure 2: **The architecture of tiny simple convolutional neural network (SCNN-Tiny).** It mainly consists of our well-designed SCNN block. In addition, we design a variant SCNN block for downsampling instead of introducing a convolution with stride 2 as patch merging for downsampling in ConvNeXt (Liu et al., 2022).

**Transformer-based Architectures.** Transformers (Vaswani et al., 2017) have made significant breakthroughs in computer vision tasks. ViT (Dosovitskiy et al., 2021) first introduces a pure Transformer architecture for visual representations. Swin (Liu et al., 2021b) utilizes window-based multi-head self-attention (MSA) for effective feature extraction. Pyramid hierarchical structure is designed in PVT (Wang et al., 2021) to extract spatial features at different scales. Depthwise convolutions are inserted in the basic transformer blocks of LocalViT (Li et al., 2021) and CPVT (Chu et al., 2023) to enhance local context. However, compared to CNNs and MLPs, ViTs face hardware compatibility limitations that restrict their wider application (Zhang et al., 2023a).

## 3  METHOD

### 3.1  OVERALL ARCHITECTURE

The overall architecture of our proposed simple convolutional neural network (SCNN) is shown in Fig 2. Assume the size of the input image is $H \times W \times 3$, we first leverage $3 \times 3$ convolution layer with stride 2 to obtain $\frac{H}{2} \times \frac{W}{2}$ feature maps, and the dimension of the feature maps is $C$ (In SCNN-Tiny, $C = 32$). We build a hierarchical representation with four stages. In the $i^{th}$ stage, we stack $N_i$ SCNN blocks (In SCNN-Tiny, $N_1 = 4, N_2 = 8, N_3 = 22, N_4 = 4$), i.e., SCNN blocks. We apply downsampling operations in the block at the end of each stage to reduce the resolution of the feature maps to half of the original. Therefore, the output feature maps of the $i^{th}$ stage is $\frac{H}{2^{i+1}} \times \frac{W}{2^{i+1}}$. We stack more $3 \times 3$ convolutions in one SCNN block and design a thinner and deeper architecture compared with ConvNeXt (Liu et al., 2022), to enlarge the receptive field. We also propose Global Sigmoid Linear Unit (GSiLU) activation function to capture spatial information in a global manner.

### 3.2  SIMPLE CONVOLUTIONAL NEURAL NETWORK BLOCK

In this section, we design the SCNN block, which uses more $3 \times 3$ spatial convolutions. As shown in Fig. 2, we design two types of SCNN blocks. One is a common block, and another is equipped with an additional downsampling operation. We design the SCNN block as follows step by step:

(1) We first apply $3 \times 3$ depthwise convolution for input features to capture spatial information.

(2) The input features are added to the output of step 1, which is a commonly used residual learning.

(3) The output feature of step 2 passes through Layer Normalization (LN), Pointwise convolution (PW), Batch Normalization (BN), Sigmoid Linear Unit (SiLU), $3 \times 3$ depthwise convolution, Batch Normalization (BN), Global Sigmoid Linear Unit (GSiLU), pointwise convolution, and batch normalization to further capture more diverse visual cues.

Table 1: Inp. Reso. and Rece. Field are the abbreviation of the Input Resolution and the Receptive Field. SCNN-Tiny is the baseline model, the same as W512, which means the dimensions are set to 64, 128, 256, and 512 respectively. We reduce the block number of all stages proportionately to design three heavy and shallow models with similar FLOPs.

| model | FLOPs | Stage 1 | | | Stage 2 | | |
|---|---|---|---|---|---|---|---|
| | | Inp. Reso. | Num | Rece. Field | Inp. Reso. | Num | Rece. Field |
| W960 | 5.1G | | 2 | 15×15 | | 2 | 15×15 |
| W768 | 4.7G | 112×112 | 3 | 23×23 | 56×56 | 4 | 31×31 |
| W640 | 4.5G | | 4 | 31×31 | | 5 | 39×39 |
| W512 | 4.5G | | 6 | 47×47 | | 8 | 63×63 |

(4) The output features of step 2 and step 3 are added together to enhance network expressiveness and alleviate the gradient vanishing. As for the SCNN block with an additional downsampling operation, the output feature of step 2 will go through 3×3 depthwise convolution with stride 2, batch normalization, pointwise convolution, and batch normalization, and then be added with the features of step 3.

The SCNN block achieves a large receptive field by stacked 3×3 convolutions and avoids the issues brought by large convolution kernels, such as the extra time in training and deployment.

The receptive field of two 3×3 convolutions are the same as one 5×5 convolution (Zhang et al., 2023a), so our design can reduce the difficulty of training and deployment brought about by the use of many large convolution kernels, and still remain large receptive field information.

### 3.3 THIN AND DEEP ARCHITECTURE

Inceptionv3 (Szegedy et al., 2016) points out that a large kernel convolution could be replaced by a multi-layer network with fewer parameters, and its experimental results prove this thought. Motivated by it, we design a thin and deep model with more 3×3 convolution instead of a heavy and shallow model with large kernel convolution. As shown in Table 1, we designed four tiny models with different depths and widths. In ImageNet dataset (Krizhevsky et al., 2012), the inputs for stage one and stage two are 112×112 and 56×56, respectively. The receptive field of the deepest model W512 is even triple the size of the shallowest one W960. Notably, the receptive field of W512 in stage two is larger than the input resolution of the feature map, which means it has a global receptive field, while other shallow models only have a local one.

### 3.4 GLOBAL SIGMOID LINEAR UNIT

Sigmoid Linear Unit (SiLU) is a widely used activation function, which was originally coined in GELU (Hendrycks & Gimpel, 2016), and later works (Ramachandran et al., 2018; Elfwing et al., 2018) demonstrate its effectiveness. After GPT using GELU, many subsequent models follow it by default, including recent ViTs (Liu et al., 2021b) and MLPs (Tolstikhin et al., 2021). GELU can be approximated as,

$$GELU(x) = x \times \Phi(x) \approx 0.5 \times x \times (1 + \tanh(\sqrt{2/\pi}) \times (x + 0.044715 \times x^3)), \quad (1)$$

where $\Phi$ means the Cumulative Distribution Function for Gaussian Distribution. Another approximate formula of GELU is:

$$GELU(x) \approx x \times \sigma(1.702 \times x), \quad (2)$$

where $\sigma$ is a sigmoid function. Similarly, Swish (Ramachandran et al., 2018) proposes to leverage automatic search techniques to discover a new activation function named Swish, which can be formulated as,

$$Swish(x) = x \times \sigma(\beta \times x). \quad (3)$$

It is easy to find that Swish has a similar formulation of GELU. The difference is that the learnable parameter in Swish is set to a fixed value of 1.702. Meanwhile, in reinforcement learning, to achieve

Table 2: Comparison with other SOTA methods on ImageNet-1K classification.

| Method | Family | Param | FLOPs | Top-1 | Throughput |
|---|---|---|---|---|---|
| Swin-T (Liu et al., 2021b) | Trans | 29M | 4.5G | 81.3 | 758 |
| Swin-S (Liu et al., 2021b) | Trans | 50M | 8.7G | 83.0 | 437 |
| Swin-B (Liu et al., 2021b) | Trans | 88M | 15.4G | 83.5 | 287 |
| HiViT-T (Zhang et al., 2023b) | Trans | 19M | 4.6G | 82.1 | 850 |
| HiViT-S (Zhang et al., 2023b) | Trans | 38M | 9.1G | 83.5 | 436 |
| HiViT-B (Zhang et al., 2023b) | Trans | 66M | 15.9G | 83.8 | 286 |
| AS-MLP-T (Lian et al., 2022) | MLP | 28M | 4.4G | 81.3 | 864 |
| AS-MLP-S (Lian et al., 2022) | MLP | 50M | 8.5G | 83.1 | 478 |
| AS-MLP-B (Lian et al., 2022) | MLP | 88M | 15.2G | 83.3 | 312 |
| CycleMLP-T (Chen et al., 2022) | MLP | 28M | 4.4G | 81.3 | 611 |
| CycleMLP-S (Chen et al., 2022) | MLP | 50M | 8.5G | 82.9 | 360 |
| CycleMLP-B (Chen et al., 2022) | MLP | 88M | 15.2G | 83.4 | 216 |
| Hire-MLP-S (Guo et al., 2022) | MLP | 33M | 4.2G | 82.1 | 808 |
| Hire-MLP-B (Guo et al., 2022) | MLP | 58M | 8.1G | 83.2 | 441 |
| Hire-MLP-L (Guo et al., 2022) | MLP | 96M | 13.4G | 83.8 | 290 |
| RaMLP-T (Lai et al., 2023) | MLP | 25M | 4.2G | 82.9 | 759 |
| RaMLP-S (Lai et al., 2023) | MLP | 38M | 7.8G | 83.8 | 441 |
| RaMLP-B (Lai et al., 2023) | MLP | 58M | 12.0G | 84.1 | 333 |
| DWNet (Han et al., 2022) | CNN | 24M | 3.8G | 81.3 | 929 |
| DWNet (Han et al., 2022) | CNN | 74M | 12.9G | 83.2 | 328 |
| ConvNeXt-T (Liu et al., 2022) | CNN | 29M | 4.5G | 82.1 | 775 |
| ConvNeXt-S (Liu et al., 2022) | CNN | 50M | 8.7G | 83.1 | 447 |
| ConvNeXt-B (Liu et al., 2022) | CNN | 89M | 15.4G | 83.8 | 292 |
| RepLKNet-31B (Ding et al., 2022) | CNN | 79M | 15.3G | 83.5 | 296 |
| InceptionNeXt-T (Yu et al., 2023) | CNN | 28M | 4.2G | 82.3 | 901 |
| InceptionNeXt-S (Yu et al., 2023) | CNN | 49M | 8.4G | 83.5 | 521 |
| InceptionNeXt-B (Yu et al., 2023) | CNN | 87M | 14.9G | 84.0 | 375 |
| SLaK-T (Liu et al., 2023) | CNN | 30M | 5.0G | 82.5 | - |
| SLaK-S (Liu et al., 2023) | CNN | 55M | 9.8G | 83.8 | - |
| SLaK-B (Liu et al., 2023) | CNN | 95M | 17.1G | 84.0 | - |
| SCNN-T (ours) | CNN | 23M | 4.5G | 83.2 | 789 |
| SCNN-S (ours) | CNN | 44M | 8.7G | 83.8 | 451 |
| SCNN-B (ours) | CNN | 75M | 15.4G | 84.0 | 324 |

the same goal of the output from one hidden unit in the expected energy-restricted Boltzmann machine (EE-RBM), SiLU (Elfwing et al., 2018) proposes an activation function for neural network function approximation:

$$SiLU(x) = x \times \sigma(x). \tag{4}$$

SiLU is a simplified version of Swish and GELU, and it does not require a learnable parameter or a fixed value inside the sigmoid function. However, SiLU computes results in all positions individually. It is unable to capture spatial information. We hope it achieves a global receptive field to let our SCNN closer to those large kernel CNNs. Thus, we propose a Global Sigmoid Linear Unit (GSiLU) activation function to capture global spatial visual cues. The formula is as follows:

$$GSiLU(x) = x \times \sigma(GAP(x)), \tag{5}$$

where GAP is a global average pooling operation. It embeds global information of every channel into a single value to produce the importance of these channels.

## 3.5 ARCHITECTURE VARIANTS

We set different numbers of blocks in Stage $1 \sim 4$ as $\{S_1, S_2, S_3, S_4\}$, and expand the channel dimensions as shown in Figure 2 to obtain variants of SCNN architecture. By balancing the performance and inference time, we designed three versions of our models, including SCNN-Tiny, SCNN-Small, and SCNN-Base. The architecture hyper-parameters of our models are:

Table 3: COCO val2017 object detection results using various backbones employing a 3x training schedule.

| Backbone | $AP^b$ | $AP^b_{50}$ | $AP^b_{75}$ | $AP^m$ | $AP^m_{50}$ | $AP^m_{75}$ | Params | FLOPs |
|---|---|---|---|---|---|---|---|---|
| Mask R-CNN (3×) | | | | | | | | |
| ResNet50 (He et al., 2016) | 41.0 | 61.7 | 44.9 | 37.1 | 58.4 | 40.1 | 44M | 260G |
| PVT-S (Wang et al., 2021) | 43.0 | 65.3 | 46.9 | 39.9 | 62.5 | 42.8 | 44M | 245G |
| AS-MLP-T (Lian et al., 2022) | 46.0 | 67.5 | 50.7 | 41.5 | 64.6 | 44.5 | 48M | 260G |
| Hire-MLP-S (Guo et al., 2022) | 46.2 | 68.2 | 50.9 | 42.0 | 65.6 | 45.3 | - | 256G |
| Swin-T (Liu et al., 2021b) | 46.0 | 68.2 | 50.2 | 41.6 | 65.1 | 44.9 | 48M | 267G |
| ConvNeXt-T (Liu et al., 2022) | 46.2 | 67.9 | 50.8 | 41.7 | 65.0 | 44.9 | 48M | 267G |
| SCNN-T (**ours**) | **47.1** | **70.2** | **54.2** | **43.7** | **67.4** | **47.2** | 42M | 252G |
| ResNet101 (He et al., 2016) | 42.8 | 63.2 | 47.1 | 38.5 | 60.1 | 41.3 | 63M | 336G |
| PVT-Medium (Wang et al., 2021) | 44.2 | 66.0 | 48.2 | 40.5 | 63.1 | 43.5 | 64M | 302G |
| AS-MLP-S (Lian et al., 2022) | 47.8 | 68.9 | 52.5 | 42.9 | 66.4 | 46.3 | 69M | 346G |
| Hire-MLP-B (Guo et al., 2022) | 48.1 | 69.6 | 52.7 | 43.1 | 66.8 | 46.7 | - | 335G |
| Swin-S (Liu et al., 2021b) | 48.5 | 70.2 | 53.5 | 43.3 | 67.3 | 46.6 | 69M | 359G |
| SCNN-S (**ours**) | **49.5** | **70.9** | **54.3** | **43.9** | 67.3 | **47.3** | 63M | 334G |
| PVT-L (Wang et al., 2021) | 44.5 | 66.0 | 48.3 | 40.7 | 63.4 | 43.7 | 81M | 364G |
| Swin-B (Liu et al., 2021b) | 48.5 | 69.8 | 53.2 | 43.4 | 66.8 | 46.9 | 107M | 496G |
| SCNN-B (**ours**) | **49.8** | **71.2** | **54.5** | **44.3** | **67.8** | **47.5** | 94M | 484G |

- SCNN-Tiny: $C = 64$, block numbers = $\{6, 8, 20, 4\}$, expand ratio = 4

- SCNN-Small: $C = 80$, block numbers = $\{8, 12, 22, 6\}$, expand ratio = 4

- SCNN-Base: $C = 96$, block numbers = $\{8, 15, 32, 6\}$, expand ratio = 4

The parameters (model size), FLOPs (computation complexity), and top-1 accuracy on ImageNet-1K of the variants of SCNN architecture are shown in Table 2.

## 4 EXPERIMENTS

In this section, starting with the evaluation of SCNN on the ImageNet-1K dataset (Deng et al., 2009) for image classification, we subsequently expand our assessment on the MS-COCO (Lin et al., 2014) object detection and instance segmentation, as well as the ADE20K (Zhou et al., 2019) semantic segmentation.

### 4.1 IMAGENET-1K CLASSIFICATION

**Experimental Setup.** To evaluate the effectiveness of our SCNN, we utilize the ImageNet-1K (Deng et al., 2009) dataset, which consists of 1.2 million training images and 50,000 validation images across 1,000 categories. Our primary metric for experimentation is the top-1 accuracy. During the training phase, we employ the AdamW optimizer with a batch size of $1024$ and initialize the learning rate at $0.001$. To facilitate learning, we incorporate cosine decay and introduce a weight decay of 0.05. The training process spans 300 epochs, with a warm-up strategy implemented for the initial 20 epochs. For data augmentation and regularization, we adopt the same strategies as ConvNeXt (Liu et al., 2022).

**Comparison with SOTA Models.** Table 2 compares SCNNs with state-of-the-art CNNs, MLPs and ViTs. Our methods demonstrate superior performance compared to Swin-Transformer (Liu et al., 2021b) and RepLKNet-31B (Ding et al., 2022). Particularly, our SCNN-T achieves a higher top-1 accuracy of 83.2% (compared to 82.5%) with fewer FLOPs (4.5G versus 5.0G) compared to SLaK-T. Additionally, our compact version of SCNN achieves better results than ConvNeXt-S (Liu et al., 2022) while requiring only approximately 50% of the computational resources. Compared with HiViT (Zhang et al., 2023b), our base version also achieves better accuracy (84.0% vs. 83.8%) with fewer FLOPs (15.4G vs. 15.9G). Compared with recent SoTA MLP, all SCNN versions are comparable with RaMLP (Lai et al., 2023).

Table 4: The semantic segmentation results of different backbones on the ADE20K validation set.

| Method | Backbone | val MS mIoU | Params | FLOPs |
|---|---|---|---|---|
| DANet (Fu et al., 2019a) | ResNet-101 (He et al., 2016) | 45.2 | 69M | 1119G |
| DeepLabv3+ (Chen et al., 2018) | ResNet-101 (He et al., 2016) | 44.1 | 63M | 1021G |
| ACNet (Fu et al., 2019b) | ResNet-101 (He et al., 2016) | 45.9 | - | - |
| DNL (Yin et al., 2020) | ResNet-101 (He et al., 2016) | 46.0 | 69M | 1249G |
| OCRNet (Yuan et al., 2020) | ResNet-101 (He et al., 2016) | 45.3 | 56M | 923G |
| UperNet (Xiao et al., 2018) | ResNet-101 (He et al., 2016) | 44.9 | 86M | 1029G |
| OCRNet (Yuan et al., 2020) | HRNet-w48 (Sun et al., 2019) | 45.7 | 71M | 664G |
| DeepLabv3+ (Chen et al., 2018) | ResNeSt-101 (Zhang et al., 2022) | 46.9 | 66M | 1051G |
| DeepLabv3+ (Chen et al., 2018) | ResNeSt-200 (Zhang et al., 2022) | 48.4 | 88M | 1381G |
| UperNet (Xiao et al., 2018) | Swin-T (Liu et al., 2021b) | 45.8 | 60M | 945G |
| | AS-MLP-T (Lian et al., 2022) | 46.5 | 60M | 937G |
| | ConvNeXt-T (Liu et al., 2022) | 46.7 | 60M | 939G |
| | Hire-MLP-S (Guo et al., 2022) | 47.1 | 63M | 930G |
| | InceptionNeXt-T (Yu et al., 2023) | 47.9 | 56M | 933G |
| | SCNN-T (ours) | **48.4** | 54M | 938G |
| UperNet (Xiao et al., 2018) | Swin-S (Liu et al., 2021b) | 49.5 | 81M | 1038G |
| | AS-MLP-S (Lian et al., 2022) | 49.2 | 81M | 1024G |
| | ConvNeXt-S (Liu et al., 2022) | 49.6 | 82M | 1027G |
| | Hire-MLP-B (Guo et al., 2022) | 49.6 | 88M | 1011G |
| | InceptionNeXt-S (Yu et al., 2023) | 50.0 | 78M | 1020G |
| | SCNN-S (ours) | **50.1** | 75M | 1025G |
| UperNet (Xiao et al., 2018) | Swin-B (Liu et al., 2021b) | 49.7 | 121M | 1188G |
| | AS-MLP-B (Lian et al., 2022) | 49.5 | 121M | 1166G |
| | ConvNeXt-B (Liu et al., 2022) | 49.9 | 82M | 1170G |
| | Hire-MLP-L (Guo et al., 2022) | 49.9 | 122M | 1125G |
| | InceptionNeXt-B (Yu et al., 2023) | **50.6** | 115M | 1159G |
| | SCNN-B (ours) | 50.1 | 108M | 1169G |

## 4.2 OBJECT DETECTION ON COCO

**Experimental Setup.** We conduct object detection employing Mask-RCNN as the framework. MS-COCO (Lin et al., 2014) dataset is selected, with 118k training data and 5k validation data. We compare SCNN with other backbones. All Hyperparameters align with Swin Transformer: AdamW optimizer, learning rate of 0.0001, weight decay of 0.05, and batch size of 2 images/GPU (8 GPUs). We use a multi-scale training strategy. Backbones are initialized with ImageNet-1K pre-trained weights. Models are trained for 36 epochs with a 3x schedule.

**Detection Results.** The performance of our SCNN on the COCO dataset is presented in Table 3, along with other architectures. Our proposed SCNN achieves superior results to the Swin Transformer and requires fewer FLOPs. Specifically, Mask R-CNN + Swin-S achieves an $AP^b$ of 48.5 with 359 GFLOPs, whereas Mask R-CNN + SCNN-S achieves an $AP^b$ of 49.5 with 334 GFLOPs.

## 4.3 SEMANTIC SEGMENTATION ON ADE20K

**Experimental Setup.** We evaluate our methods on ADE20K (Zhou et al., 2019), a challenging semantic segmentation dataset. We use the efficient UperNet (Xiao et al., 2018) framework. In training, we initialize the backbone with ImageNet weights and use Xavier initialization for other layers. AdamW optimizer with initial learning rate $1.0 \times 10^{-4}$ is used. Training involves 160k iterations, batch size 16 on 8×A100 GPUs, weight decay 0.01, and polynomial decay schedule with power 0.9. Data augmentation includes random horizontal flipping, rescaling (0.5-2.0), and photometric distortion. The stochastic depth ratio is set to 0.3. During training, images are randomly resized and cropped to $576 \times 576$, and during testing, they are rescaled to have a shorter side of 576 pixels. The evaluation metric is multi-scale mean Intersection over Union (MS mIoU).

Table 5: Ablation analysis on the convolution in SCNN block and the GSiLU. PreConv means the first convolution, and MidConv means the second convolution in the block.

| PreConv | MidConv | GSiLU | Top-1 (%) | Param | FLOPs |
|:---:|:---:|:---:|:---:|:---:|:---:|
| ✓ | ✓ | ✓ | **83.2** | 23M | 4.5G |
| ✓ | ✓ | | 83.0 | 23M | 4.5G |
| ✓ | | | 82.7 | 23M | 4.4G |
| | ✓ | | 82.6 | 23M | 4.4G |

Table 6: Ablation analysis on the model depth with similar complexity. Block numbers mean the numbers in four stages, while channel dims mean the channel dimensions in same four stages.

| block numbers | channel dims | Params | FLOPs | Top-1 |
|:---:|:---:|:---:|:---:|:---:|
| 2,3,5,1 | 120.240,480,960 | 21M | 5.1G | 82.1 |
| 3,4,8,2 | 96,192,384,768 | 23M | 4.7G | 82.5 |
| 4,5,12,3 | 80,160,320,640 | 24M | 4.5G | 82.9 |
| 6,8,20,4 | 64,128,256,512 | 23M | 4.5G | **83.2** |

**Segmentation Result.** Table 4 presents a performance comparison between our SCNN and state-of-the-art architectures on the ADE20K dataset. Despite having similar FLOPs, SCNN-T achieves superior results compared to Swin-T, with an MS mIoU of 48.4 versus 45.8.

### 4.4 ABLATION STUDY

The main component of the SCNN is two depthwise convolutional layers and SiLU with global average pooled features, which directly affects the receptive field. Besides, the settings of model depth also contribute a lot to the receptive field and non-linear fitting capability. We conduct ablation experiments to verify these factors. Unless otherwise stated, all experiments are based on SCNN-T on ImageNet-1K (Krizhevsky et al., 2012).

**The Impact of GSiLU.** As shown in Table 5 line two, when we remove the GSiLU and adopt a traditional SiLU function, the result gets a decrease of 0.2%. This proves the importance of capturing long-range visual cues because SiLU only uses original feature maps as the input, which can not capture any spatial information, while GSiLU could capture global spatial information.

**The Impact of SCNN Block.** As shown in Table 5 lines three and four, the result is markedly declined when we remove one convolutional layer in the SCNN block. The receptive field will become almost halved by using only one convolution in a block. Thus, SCNN will degenerate into a traditional small kernel CNN like MobileNetV2 (Sandler et al., 2018).

**The Impact of Model Depth.** In table 6, besides SCNN-T, we design another three models with different depths. We found that a thinner, deeper architecture could obtain better results than heavier and shallower models. Surprisingly, the shallowest model with more FLOPs even gets a -1.1% performance compared with SCNN-T. The main reasons may a smaller receptive field and worse non-linear fitting capability.

## 5 CONCLUSION

We propose the Simple Convolutional Neural Network (SCNN) that mainly employs a sequence of stacked 3×3 convolutions to capture visual cues of various sizes. Though the architecture is simple, SCNN surpasses the state-of-the-art CNNs with larger kernels. SCNN is a thin and deep model, encouraging more layers of convolutions to capture more spatial information under the same computing complexity. Furthermore, we introduce the innovative SCNN block comprising two 3×3 depthwise convolutions to enlarge the receptive field further. We also replace the input of the Sigmoid Linear Unit (SiLU) activation function with global average pooled features to capture global information. Experimental results on the downstream tasks (object detection, semantic segmentation, and instance segmentation) further verify the superiority of SCNN.

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

## A  APPENDIX

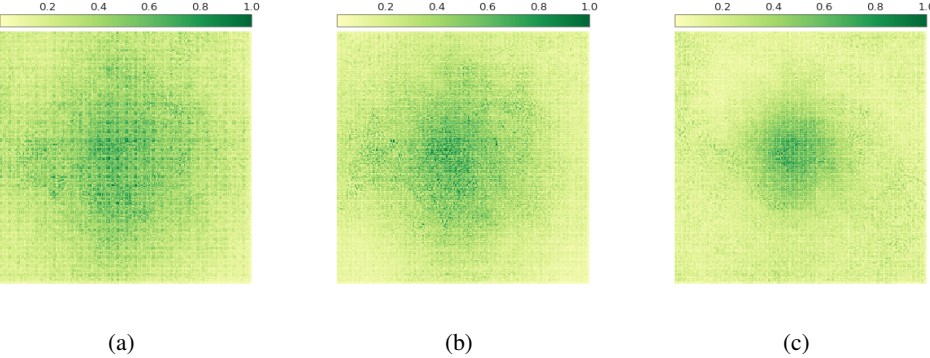

(a)  (b)  (c)

Figure 3: The Effective Receptive Field (ERF). a) only use the second convolution in the block; b) use both the first and second convolutions in the block; c) use two convolutions and GSiLU activation functions.

As shown in Figure 3, models with these three kinds of configures can obtain large ERF. The latter two models are more discriminative. The basic model with only one convolution has similar scores in all positions, while the SCNN model focuses more on center and edge areas.

