# OpenReview forum: "Simple CNN for Vision"
_ICLR.cc/2024/Conference — Submitted to ICLR 2024_

### Official Review · Reviewer_tQCD · 2023-10-28

**Soundness:** 3 good
**Presentation:** 3 good
**Contribution:** 3 good
**Rating:** 6
**Confidence:** 1

**Summary:**

The paper provides a clear background on the significance of large kernel convolutions in the current landscape of CNNs. The authors also clearly demonstrated their motivations for SCNNs and showed that a sequence of stacked 3×3 convolutions surpasses state-of-the-art CNNs utilizing larger kernels. The introduction provides context, discusses the trend towards large kernels, and then sets the stage for their proposition.

**Strengths:**

1. The approach of using a sequence of stacked 3×3 convolutions to surpass state-of-the-art CNNs employing larger kernels is innovative.

2. The technical sounds solids, including the actual implementation specifics of SCNN, and corresponding theoretical explanations.

**Weaknesses:**

1. Experiments: The paper should delve deeper into the experimental setup, data augmentation techniques, training specifics, and more. For example, it is unclear to compare with other baselines that have different parameter sizes. The author should clarify this section.

2. Lack of Visualizations: Additional figures visualizing feature maps or demonstrating how the receptive field increases would offer more insights into the workings of SCNN.

**Questions:**

Please refer to the weakness, W1

---

> ### Author Response · Authors · 2023-11-22
> **Response to Reviewer tQCD**
>
> We thank the reviewer for the constructive feedback. We make point-to-point clarification for your concerns as follows:
>
> **Q1:** The paper should delve deeper into the experimental setup, data augmentation techniques, training specifics, and more. For example, it is unclear to compare with other baselines that have different parameter sizes. The author should clarify this section.
>
> **A1:** Thanks for your comment. All of our settings are the same as Swin Transformer and ConvNeXt, we have not made any further adjustments.
>
> **Q2:** Lack of Visualizations: Additional figures visualizing feature maps or demonstrating how the receptive field increases would offer more insights into the workings of SCNN.
>
> **A2:** We use the script in RepLKNet(https://github.com/DingXiaoH/RepLKNet-pytorch/tree/main/erf) to visualize the receptive field, and put these images in the **appendix (page 12)**. For models trained in imagenet-1k, all of them could capture a large receptive field. After adding more convolutions and GSiLU, the features become more discriminative and the final model focuses more on center and edge areas.

---

### Official Review · Reviewer_of72 · 2023-10-30

**Soundness:** 3 good
**Presentation:** 2 fair
**Contribution:** 2 fair
**Rating:** 3
**Confidence:** 4

**Summary:**

This paper proposes a Simple Convolutional Neural Network (SCNN) that uses only 3x3 convolutions but outperforms models with larger kernels. The main ideas are: (1) Thin and deep architecture with more 3x3 layers to capture spatial information under compute constraints. (2) Stacking two 3x3 depthwise convolutions to enlarge receptive field. (3) Using global average pooling in activation (GSiLU) to capture global information. The model is evaluated on ImageNet classification, COCO detection/segmentation, and ADE20K segmentation, achieving improved results.

**Strengths:**

The paper comprehensively validates the effectiveness and efficiency of the proposed SCNN architecture. It outperforms models with larger kernels like ConvNeXt and SLaK on ImageNet classification by up to 0.7% top-1 accuracy with less FLOPs (Table 2). For dense prediction tasks, SCNN as a backbone for Mask R-CNN improves COCO object detection AP by 0.9% over Swin Transformer while requiring lower computation (Table 3). Similarly, SCNN exceeds Swin Transformer by 2.6% mIoU on ADE20K semantic segmentation with comparable FLOPs (Table 4). Detailed ablation studies demonstrate the impact of key components like the thin and deep architecture, double 3x3 convolutions, and global context modeling with GSiLU.

**Weaknesses:**

•	The paper lacks some analysis on how the proposed architecture captures spatial context and increases receptive field size. The introduction describes this as a key motivation, but there is little discussion in the experiments. Some visualization or measurements of the receptive field size could provide more insight.
•	The paper does not discuss in detail some other related work on improving convolutional backbones, such as [1,2]. Comparing and contrasting with these methods could highlight the novelty of SCNN's approach.
[1] Meng-Hao Guo, Cheng-Ze Lu, Zheng-Ning Liu, Ming-Ming Cheng, Shi-Min Hu:Visual attention network. Comput. Vis. Media 9(4): 733-752 (2023)
[2] Wenhai Wang, Enze Xie, Xiang Li, Deng-Ping Fan, Kaitao Song, Ding Liang, Tong Lu, Ping Luo, Ling Shao: PVT v2: Improved baselines with Pyramid Vision Transformer. Comput. Vis. Media 8(3): 415-424 (2022)
•	Beyond incremental improvements on established benchmarks, in-depth insights or observations about the properties of the SCNN architecture could increase the depth of the contribution.

**Questions:**

For global context modeling, how does GSiLU compare to other approaches like SENet? I think the global context modeling in SCNN is not very different from widely used techniques like SE modules（Channel Attention for Convolution）. The paper could benefit from more comparison and discussion about the relative merits of GSiLU.

Some other concerns are in Weaknesses.

---

> ### Author Response · Authors · 2023-11-22
> **Response to Reviewer of72**
>
> We sincerely appreciate the time and efforts you have dedicated to reviewing our paper.  We thank your suggestions and make point-to-point clarification to the weaknesses and your questions as follows:
>
> **Q1:** The paper lacks some analysis on how the proposed architecture captures spatial context and increases receptive field size. The introduction describes this as a key motivation, but there is little discussion in the experiments. Some visualization or measurements of the receptive field size could provide more insight.
>
> **A1:** We use the script in RepLKNet(https://github.com/DingXiaoH/RepLKNet-pytorch/tree/main/erf) to visualize the receptive field, and put these images in the **appendix (page 12)**. For models trained in imagenet-1k, all of them could capture a large receptive field. After adding more convolutions and GSiLU, the features become more discriminative and the final model focuses more on center and edge areas.
>
> **Q2:** The paper does not discuss in detail some other related work on improving convolutional backbones, such as [1,2]. Comparing and contrasting with these methods could highlight the novelty of SCNN's approach. [1] Meng-Hao Guo, Cheng-Ze Lu, Zheng-Ning Liu, Ming-Ming Cheng, Shi-Min Hu:Visual attention network. Comput. Vis. Media 9(4): 733-752 (2023) [2] Wenhai Wang, Enze Xie, Xiang Li, Deng-Ping Fan, Kaitao Song, Ding Liang, Tong Lu, Ping Luo, Ling Shao: PVT v2: Improved baselines with Pyramid Vision Transformer. Comput. Vis. Media 8(3): 415-424 (2022) • Beyond incremental improvements on established benchmarks, in-depth insights or observations about the properties of the SCNN architecture could increase the depth of the contribution.
>
> **A2:** We cite and compare with PVT in instance segmentation tasks, but we overlook PVT v2 and VAN because they are published in a newly established journal (2015) we overlook. We will cite them in the final version. Compared with PVT v2, SCNN-T gets **the same accuracy (83.2%) with only 65% FLOPs (4.5G vs 6.9G)**. Compared with VAN, SCNN-T obtains **better results (83.2% vs 82.8%) with fewer FLOPs (4.5G vs 5.0G)**.
>
> **Q3:** For global context modeling, how does GSiLU compare to other approaches like SENet? I think the global context modeling in SCNN is not very different from widely used techniques like SE modules（Channel Attention for Convolution）. The paper could benefit from more comparison and discussion about the relative merits of GSiLU.
>
> **A3:** However, both GSiLU and SE belong to gated activation functions. And replacing GSiLU with SE, the performance gain is marginal (from 83.2% to 83.3). For a fair comparison, we utilize GSiLU because both SiLU and GSiLU are parameter-free activation functions, and we only add a global average pooling. SE requires an extra MLP, which introduces a large number of parameters.

---

### Official Review · Reviewer_SZ8J · 2023-10-31

**Soundness:** 2 fair
**Presentation:** 1 poor
**Contribution:** 4 excellent
**Rating:** 6
**Confidence:** 4

**Summary:**

In the provided submission, the authors present a Simple Convolutional Neural Network (SCNN) that using only 3x3 depthwise convolutions, outperforms CNNs that employ larger kernels. A notable enhancement is the incorporation of global average pooled features into the Sigmoid Linear Unit (SiLU) activation function, the paper named it GSiLU, which enables these small kernels to capture comprehensive spatial information.

**Strengths:**

1) The paper effectively challenges the notion that larger kernels are the way for CNN advancements.
2) The presented model is more lightweight than its counterparts, achieving comparable performance with the same FLOPs.
2) The authors have conducted a robust suite of experiments to validate their model.

**Weaknesses:**

1) The paper could benefit from a more polished and professional tone in its presentation and writing style.
2) Some of the claims or hypotheses put forth lack empirical validation through experiments, which could strengthen the paper's assertions.
3) While it's understandable given potential computational constraints, the model was not trained on ImageNet 21K, a limitation that might affect generalization claims.
4) The paper doesn't explore the performance of larger-sized versions of their model. Although this might be due to resource limitations, such exploration could provide additional insights into the model's scalability and robustness.

**Questions:**

1) Regarding Table 5's ablation analysis on the SCNN block, it is unclear why the number of parameters and FLOPs remain constant when PreConv and MidConv are removed. Could the authors clarify if there were any mechanisms employed to maintain these metrics, and if so, elaborate on the methodology used?
2) The paper describes the use of both Layer Normalization (LN) and Batch Normalization (BN) within the architecture, with LN explicitly employed as the initial normalization layer. Could the authors explain the reason behind this specific arrangement? Furthermore, it would be beneficial if the authors could provide an ablation study examining the impact of the positioning of these normalization layers within the network.
3) The paper describes the use of both SiLU and GSiLU activation functions within the architecture, with the use of SiLU as the first activation layer. Could the authors provide insight into the reason underpinning this specific sequence?
4) The claim that GSiLU enhances the performance of small 3x3 kernels by capturing global spatial information warrants empirical validation. Could the authors clarify if GSiLU doesn't have similar benefits to larger kernel sizes?
5) The functionality of GSiLU, which zeros out channels with sufficiently negative averages, prompts a request for statistical analysis. Could the authors provide data on the proportion of channels that are effectively zeroed by the GSiLU activation in practice?
6) As part of a review process, I believe it is expected to validate the findings through an independent examination of the code and model (Hence, my low confidence score.). To maintain the integrity of my review, I request access to the relevant code and model. This will enable me to verify the results personally.

---

> ### Author Response · Authors · 2023-11-22
> **Response to Reviewer SZ8J**
>
> We sincerely appreciate your comprehensive review and positive feedback on our work. Below, we address the weaknesses and questions.
>
> **Weaknesses**
>
> **Q1:** The paper could benefit from a more polished and professional tone in its presentation and writing style.
>
> **A1:** Thank you, we will polish the manuscript.
>
> **Q2:** Some of the claims or hypotheses put forth lack empirical validation through experiments, which could strengthen the paper's assertions.
>
> **A2:** Thank you, we will clearly check our claims or hypotheses and provide more references, more experiments, and more visualizations in the final version.
>
> **Q3:** While it's understandable given potential computational constraints, the model was not trained on ImageNet 21K, a limitation that might affect generalization claims.
>
> **A3:** ImageNet 21K is too large for our limited resources. We need more time to ask for help from technology companies to do this experiment in the final version.
>
> **Q4:** The paper doesn't explore the performance of larger-sized versions of their model. Although this might be due to resource limitations, such exploration could provide additional insights into the model's scalability and robustness.
>
> **A4:** Due to the limitation of resources, we train the base version of the model (15.4G). We will try to ask for help from some technology companies, to train a larger version (e.g. 50G FLOPs).
>
> **Questions**
>
> **Q5:** Regarding Table 5's ablation analysis on the SCNN block, it is unclear why the number of parameters and FLOPs remain constant when PreConv and MidConv are removed. Could the authors clarify if there were any mechanisms employed to maintain these metrics, and if so, elaborate on the methodology used?
>
> **A5:** We use the THOP (https://github.com/Lyken17/pytorch-OpCounter) to measure the FLOPs. Both PreConv and MidConv are depthwise convolutions. The computing complexity of them is less than 0.1G FLOPs, thus, reduced FLOPs are ignored due to round-up and round-down. For example, both 4.549G and 4.451G are represented as 4.5G.
>
> **Q6:** The paper describes the use of both Layer Normalization (LN) and Batch Normalization (BN) within the architecture, with LN explicitly employed as the initial normalization layer. Could the authors explain the reason behind this specific arrangement? Furthermore, it would be beneficial if the authors could provide an ablation study examining the impact of the positioning of these normalization layers within the network.
>
> **A6:** To train the model faster, we use apex in PyTorch to train an fp16-fp32 mixed model. We find removing the LN will lead to a numeric overflow problem in the base version (15.4G). In tiny version (4.5G) and small version (8.7G), with or without LN, the result remains unchanged.
>
> **Q7:** The paper describes the use of both SiLU and GSiLU activation functions within the architecture, with the use of SiLU as the first activation layer. Could the authors provide insight into the reason underpinning this specific sequence?
>
> **A7:** SiLU has a similar formulation to GELU and Swish, which are widely used in most modern backbones. Thus, it is our default setting to use SiLU in all positions. And, we find replacing the second SiLU with GSiLU will obtain a better result (+0.1%) compared with replacing the first one.
>
> **Q8:** The claim that GSiLU enhances the performance of small 3x3 kernels by capturing global spatial information warrants empirical validation. Could the authors clarify if GSiLU doesn't have similar benefits to larger kernel sizes?
>
> **A8:** We use the script in RepLKNet(https://github.com/DingXiaoH/RepLKNet-pytorch/tree/main/erf) to visualize the receptive field, and put these images in the **appendix (page 12)**. For models trained in imagenet-1k, all of them could capture a large receptive field. After adding more convolutions and GSiLU, the features become more discriminative and the final model focuses more on center and edge areas.
>
> **Q9:** The functionality of GSiLU, which zeros out channels with sufficiently negative averages, prompts a request for statistical analysis. Could the authors provide data on the proportion of channels that are effectively zeroed by the GSiLU activation in practice?
>
> **A9:** In the tiny version, the stage one block has 64 channels, we find there is no zero value in these channels, and more than 50% are between 0.3 and 0.7. We think the sigmoid function makes it difficult to generate a zero-value channel.
>
> **Q10:** As part of a review process, I believe it is expected to validate the findings through an independent examination of the code and model (Hence, my low confidence score.). To maintain the integrity of my review, I request access to the relevant code and model. This will enable me to verify the results personally.
>
> **A10:** All codes and models will be public after being accepted. If it is rejected, we will make this paper, codes, and models public in arXiv and github soon.

---

> ### Comment · Reviewer_SZ8J · 2023-12-01
> **Missing code and model for reproducing the results.**
>
> Thank you to the authors for your response and the effort put into this work.
>
> Unfortunately, the most crucial aspect for me, which is personal reproducibility of results is not met in this case. Therefore, I must lower my score, although I will increase my confidence level in recognition of your engagement and efforts.

---

### Official Review · Reviewer_gmsA · 2023-11-01

**Soundness:** 3 good
**Presentation:** 3 good
**Contribution:** 2 fair
**Rating:** 5
**Confidence:** 4

**Summary:**

This paper proposes Simple Convolution Neural Networks (SCNN) for a bunch of fundamental vision tasks (classification, detection segmentation). It conducts extensive comparison to existing improvements over CNN such as ConvNeXt, RepLKNet, and ViTs. Results shows that it could achieve superb results comparing to those SOTA CNNs and ViTs.

**Strengths:**

The presentation is clear and the idea is fairly simple.

**Weaknesses:**

In Figure 2, putting aside SILU and GSILU, the architecture of SCNN looks very similar to mobilenet and its variants; Could the authors provide results comparison to MobileNet with the same layers, depth and width, but without SILU and GSILU blocks? I am curious whether the improvement is coming from SILU or GSILU.

**Questions:**

It claims GSILU leads to rich spatial information, could you provide a receptive field size analysis/illustration when comparing network with or without SILU and GSILU?

---

> ### Author Response · Authors · 2023-11-22
> **Response to Reviewer gmsA**
>
> We sincerely appreciate your comment which will help us strengthen the manuscript. We perform point-to-point clarification for your concerns as follows:
>
> **Q1:** Could the authors provide results comparison to MobileNet with the same layers, depth, and width, but without SILU and GSILU blocks? I am curious whether the improvement is coming from SILU or GSILU.
>
> **A1:** The top-1 accuracy of MobileNetV2 with the same settings you mentioned is **81.9% under 4.5G FLOPs**. Notably, by replacing the ReLU activation with SiLU functions, it becomes **82.6% under 4.4G FLOPs**, which is recorded in Tabel 5 line4. It is clear to demonstrate the effectiveness.
>
> **Q2:** It claims GSILU leads to rich spatial information, could you provide a receptive field size analysis/illustration when comparing network with or without SILU and GSILU?
>
> **A2:** We update the paper and add a visualization of the receptive field to the **appendix (page 12)**. It is clear that for the model trained in imagenet-1k, replacing SiLU with GSiLU makes the features more discriminative, thus the final model focuses more on center and edge areas. Specifically, we use the script in RepLKNet(https://github.com/DingXiaoH/RepLKNet-pytorch/tree/main/erf) to visualize the receptive field.

---

> ### Comment · Reviewer_gmsA · 2023-11-23
>
> Thanks for providing extra results to answer my concerns.
>
> I have a follow up question for Table-5.
> According to your response, Line-4 is MobileNet-V2,  Line-2 is SCNN.
> What about MobileNet-V2 + GSILU (PreConv + GSiLU), since in Figure-2, GSiLU is not replacing BN, but combing with BN to provide effects.
>
> Besides, why do you add SiLU after first BN, but add GSiLU after 2nd BN. It is that based on experimental study?

---

> > ### Author Response · Authors · 2023-11-23
> > **Response to Reviewer gmsA**
> >
> > Thanks for your quick feedback! Yes, 81.9% is from the original MobileNet v2(ReLU), while line 4 in Table 5 is MobileNet v2(GELU). Line 1 and Line 2 are SCNN using GSiLU or SiLU. We checked our research record but found we missed the experiment MobileNet-V2 + GSILU (PreConv + GSiLU). We found that SCNN + GSiLU (PreConv+GSiLU) achieves 82.9%, and SCNN + GSiLU (MidConv+GSiLU) achieves 83.0%. Thank you for your insightful suggestion. Due to the limited time in rebuttal, we will add the experiment about MobileNet v2+GSiLU with more analysis in the final version.
> >
> > For adding SilU after the first BN and GSiLU after 2nd BN, our motivation is that GSiLU exploits global spatial information and works channel-wise; thus, applying GSiLU after 3x3 convolution (2nd Conv) can help GSiLU to utilize more spatial information, while the first convolution is 1x1. Moreover, if moving GSiLU to after the first BN, it will capture global information first and then utilize 3x3 convolution to capture local information, which may disturb the spatial information processing. Our experimental results also demonstrate that applying GSiLU after 2nd BN can bring 0.1% improvement on imagenet-1k.

---

### Meta-Review · Area_Chair_GMPY · 2023-12-05

**Metareview:**

This paper proposes Simple Convolution Neural Networks (SCNN) which only uses 3x3 depthwise convolutions, and replaces the input of the popular Sigmoid Linear Unit  activation function with global average pooled features to capture all spatial information. It achieves good performance on a bunch of fundamental vision tasks, e.g., classification, detection segmentation.

The main strength of this work is that it is very simple and efficient. The main issues of three  reviewers are the incremental novelty of this work: it does not bring a new insights , and merely combines old concepts/techniques, e.g., MobileNet-v2 combined with GSiLU. Moreover, some reviwers also worry its adequate   relevant works.  Considering most reviewers have low intentions to accept this work, we cannot accept it.

**Justification For Why Not Higher Score:**

1) The main issues of three  reviewers are the incremental novelty of this work: it does not bring a new insights , and merely combines old concepts/techniques, e.g., MobileNet-v2 combined with GSiLU.

2) Some reviwers also worry its adequate  relevant works.

**Justification For Why Not Lower Score:**

N/A

---

### Decision · Program_Chairs · 2024-01-16

Reject